# Target Enrichment Enables the Discovery of lncRNAs with Somatic Mutations or Altered Expression in Paraffin-Embedded Colorectal Cancer Samples

**DOI:** 10.3390/cancers12102844

**Published:** 2020-10-01

**Authors:** Susana Iraola-Guzmán, Anna Brunet-Vega, Cinta Pegueroles, Ester Saus, Hrant Hovhannisyan, Alex Casalots, Carles Pericay, Toni Gabaldón

**Affiliations:** 1Institute for Research in Biomedicine (IRB Barcelona), The Barcelona Institute of Science and Technology, Baldiri Reixac 10, 08028 Barcelona, Spain; susana.iraola@irbbarcelona.org (S.I.-G.); cinta.pegueroles@bsc.es (C.P.); ester.saus@irbbarcelona.org (E.S.); hrant.hovhannisyan@bsc.es (H.H.); 2Barcelona Supercomputing Centre (BSC-CNS), Jordi Girona 29, 08034 Barcelona, Spain; 3Institut d’Investigació i Innovació Parc Taulí I3PT, Parc Taulí Hospital Universitari, Universitat Autònoma de Barcelona, 08208 Sabadell, Spain; ABrunetV@tauli.cat; 4Oncology Service, Parc Taulí Hospital Universitari, Universitat Autònoma de Barcelona, 08208 Sabadell, Spain; cpericay@tauli.cat; 5Pathology Service, Parc Taulí Hospital Universitari, Universitat Autònoma de Barcelona, 08208 Sabadell, Spain; acasalots@tauli.cat; 6Catalan Institution for Research and Advanced Studies (ICREA), 08010 Barcelona, Spain

**Keywords:** lncRNAs enrichment, CRC, FFPE samples

## Abstract

**Simple Summary:**

Alterations in long noncoding RNAs and their mutations have been increasingly recognized in tumorogenesis and cancer progression awakening especial interest as potential novel cancer biomarkers and therapeutic targets. The use of adjuvant chemotherapy in stage II colorectal cancer patients is challenging, and new biomarkers are required to identify patients with high probability of relapse. We focused on translational potential of non-coding RNAs in colorectal cancer. In this study, we aim to validate a new tool which couples target enrichment and RNAseq for transcriptomics studies of lncRNAs in formalin-fixed paraffin embedded (FFPE) tissue samples. Our results show that this new approach efficiently detects lncRNAs and differences in their expression between healthy and tumor FFPE tissues, as well as somatic mutations in expressed lncRNAs, identifying novel lncRNAs as potential candidates for colorectal cancer. This new approach could represent a promising avenue that would reduce costs and enable more efficient translational research.

**Abstract:**

Long non-coding RNAs (lncRNAs) play important roles in cancer and are potential new biomarkers or targets for therapy. However, given the low and tissue-specific expression of lncRNAs, linking these molecules to particular cancer types and processes through transcriptional profiling is challenging. Formalin-fixed, paraffin-embedded (FFPE) tissues are abundant resources for research but are prone to nucleic acid degradation, thereby complicating the study of lncRNAs. Here, we designed and validated a probe-based enrichment strategy to efficiently profile lncRNA expression in FFPE samples, and we applied it for the detection of lncRNAs associated with colorectal cancer (CRC). Our approach efficiently enriched targeted lncRNAs from FFPE samples, while preserving their relative abundance, and enabled the detection of tumor-specific mutations. We identified 379 lncRNAs differentially expressed between CRC tumors and matched healthy tissues and found tumor-specific lncRNA variants. Our results show that numerous lncRNAs are differentially expressed and/or accumulate variants in CRC tumors, thereby suggesting a role in CRC progression. More generally, our approach unlocks the study of lncRNAs in FFPE samples, thus enabling the retrospective use of abundant, well documented material available in hospital biobanks.

## 1. Introduction

Long non-coding RNAs (lncRNAs) are a large and heterogeneous class of transcripts that are longer than 200 nucleotides, and have limited coding potential [1,2]. Most lncRNAs are transcribed by RNA polymerase II, capped, spliced, and polyadenylated [3]. However, compared to protein-coding mRNAs, lncRNAs are shorter, have fewer exons, are expressed at lower levels and in a more tissue- and cell-specific manner, and are alternatively spliced more frequently [4]. From an evolutionary standpoint, lncRNAs are poorly conserved, which limits their study in model organisms [5]. Accumulating evidence supports the involvement of lncRNAs in all aspects of gene regulation and, accordingly, numerous lncRNAs have been linked to diverse human diseases, including cancer [6,7]. For instance, transcriptomic analyses have shown that changes in the expression of lncRNAs are among the most commonly observed alterations in cancer [8,9]. Although the role of lncRNAs is still poorly understood, their involvement in cancer paves the way to novel therapeutic, diagnostic, or prognostic approaches centered on these molecules.

To gain further insights into the role of lncRNAs in cancer origin and progression, it is necessary to study their expression patterns in healthy tissues and in tumors at different stages of development. The application of high throughput RNA sequencing (RNA-Seq) enables the identification of novel lncRNAs and provides information on the sequence and differential expression of transcripts [10]. However, the low expression of lncRNAs hampers this type of analysis, particularly when working with preserved samples such as formalin-fixed, paraffin-embedded (FFPE) tissues. FFPE tissues are an abundant and highly valuable source of material for research purposes and are generally accompanied by substantial clinical data. However, this form of preservation leads to partial degradation and fragmentation of nucleic acids [11]. As a result, most transcriptomic studies of cancer are limited to fresh-frozen (FF) tissues, and those using FFPE samples generally rely on RT-qPCR or microarray approaches. Nevertheless, microarray-based studies have shown that, overall, the expression profiles and the inferred sets of differentially expressed genes are similar when using FFPE or FF samples [12]. More recently, studies based on RNA-Seq have confirmed this result [13], finding high correlations between expression levels measured in matched FFPE and FF samples (*r* > 0.80–0.96 depending on the study) [14,15,16,17].

Despite recent advances [18], the study of lncRNAs in FFPE samples is still in its infancy. High-density microarrays can be designed to study a specific set of pre-selected RNAs, but they present limitations in comparison to RNA-Seq technologies, such as high background noise levels and a limited dynamic range [19]. Alternative approaches based on surface or bead-linked hybridization probes can be used to enrich a desired subset of transcripts before RNA-Seq. For instance, the RNA CaptureSeq approach is efficient for diverse types of samples [20,21]. In 2015, a specific capture method for the transcriptomic analysis of degraded RNA was developed [22], which provided excellent data from clinical specimens, including FFPE and FF samples. Additionally, SureSelect^XT^ Targeted RNA (Agilent Technologies, Santa Clara, CA, USA) efficiently targets low input samples such as FFPE and allows custom selection of target regions [23]. However, most existing designs target only protein coding genes, and are thus not able to capture lncRNAs [24]. Currently, only one enrichment design that specifically targets human lncRNAs is commercially available, namely SeqCap LncRNA probe kit (Roche NimbleGen, Inc., Madison, WI, USA) [25]. However, its use on FFPE tissue has not been validated to date.

Colorectal cancer (CRC) is the third most common cause of cancer mortality worldwide. A total of 1.8 million new cases are diagnosed globally, with a 5-year prevalence of 4.7 million [26]. CRC can develop gradually through an accumulation of genetic and/or epigenetic alterations that lead to the transformation of colonic mucosa into invasive cancer [27,28]. To date, multiple lncRNAs have been identified in CRC [29] and several studies have shown their involvement in key biological processes, from transcriptional regulation [30,31,32,33], to cellular mechanisms [34,35,36,37,38], and tumor progression [39]. Some reports have shown that cancer cells secrete lncRNAs into biological fluids [40,41]. This mechanism suggests that oncogenic signals can be spread to other cells, and it opens interesting perspectives for biomarker discovery [42,43]. However, only a limited number of CRC-related lncRNAs have been characterized to date, and their functional roles remain poorly understood.

To gain further insights into the role of lncRNAs in CRC and to validate the potential of a target enrichment approach coupled to RNA-Seq using FFPE samples, we designed and tested a custom lncRNA enrichment probe set. We focused on samples from stage II CRC patients, as this is a critical phase of the disease for which novel prognostic tools are needed [44]. For patients with stage II CRC, there is no clear consensus on how to approach treatment after surgery and nowadays the decision to administer adjuvant chemotherapy is a challenge. An estimated 75–80% of people with stage II colon cancer will be cancer-free 5 years later, without adjuvant chemotherapy, but 20–25% will not. Some of these patients may benefit from having chemotherapy after surgery. However, the risk of toxicity is very high, and every case needs to be evaluated carefully. Therefore, and according to current guidelines, adjuvant chemotherapy is only administered in young patients with relapse risk factors, such as: tumor size (T4, stage II b), number of nodes assessed (nodes < 12), degree of cellular differentiation (well vs poorly differentiated), extramural lymphatic or perineural venous invasion, and intestinal perforation. Questions remain not only about which patients will benefit from chemotherapy treatment but also what type of chemotherapy is advisable to use. In this context, increased research into lncRNAs may lead to novel biomarkers for the identification of patients at risk of metastases or for the prediction of response to chemotherapy [45,46]. Our results show that this new approach efficiently detects lncRNAs and the differences in their expression between healthy and tumor FFPE tissues. In addition, the method allows the detection of somatic mutations in expressed lncRNAs. This method thus emerges as a promising new tool that would increase resolution in lncRNA transcriptomic studies, and facilitate translational research in cancer and, more generally, the study of any disease that could benefit from the analysis of FFPE samples. Finally, our study identifies novel lncRNAs with altered expression or sequence in CRC, which are promising candidates for future research.

## 2. Results and Discussion

### 2.1. Design of a lncRNA-Focused Enrichment

The landscape of annotated human lncRNAs is constantly evolving [3], and many lncRNAs recently linked to CRC do not appear in current annotations of the human genome. To maximize our coverage of lncRNAs with annotated functions and those previously found to be involved in CRC, we mined literature and database information (see Section 3). In brief, we included all annotated lncRNAs in the Ensemble database that do not overlap protein coding genes. We included additional transcripts with functional annotations or a potential role in cancer, particularly in CRC, by curating various sources, including: (i) miTranscriptome, a curated catalogue of genes with high representation of cancer-specific lncRNAs based on RNA-Seq analysis of over 6500 cancer samples [9]; (ii) Lnc2Cancer, a manually curated database from public data with more than 1000 lncRNAs associated with human cancer [47]; (iii) a recent survey of lncRNAs associated to CRC [29]; and (iv) lncRNAdb v2.0, a manually curated database including functional lncRNAS [48]. The final dataset comprised 7812 lncRNA genes, of which 216 can be considered as CRC-related on the basis of previous knowledge. We designed capture probes for SeqCap EZ Choice Enrichment kit (Roche NimbleGen, Inc., Madison, WI, USA) according to specifications (CoLong design, see Section 3).

### 2.2. Validation of Probe-Based Enrichment of lncRNAs

We validated the SeqCap enrichment approach using the CoLong design in parallel with a commercially available alternative (SeqCap-lncRNA, see Section 3), whose use on FFPE samples has not been tested previously. In addition to FFPE samples, we included two CRC cell lines (HCT116 and HT29) to compare lncRNA enrichment performance with and without polyA selection (Figure 1A), which is a standard transcript enrichment technique that is not suitable for FFPE (Appendix A). Our results (Figure 1B) show that target enrichment protocols efficiently enhanced the detection of lncRNAs, independently of the previous use of a polyadenylated (polyA) RNA capture protocol, both in terms of the number of detected lncRNAs and their coverage in transcripts per million (TPM).

Most importantly, the use of both SeqCap-lncRNA and CoLong designs enabled the detection of lncRNAs in FFPE samples, starting from as little as 0.1–0.15 µg of partially degraded RNA (RNA integrity number (RIN )average: 2.20 ± 0.33), detecting a similar number of transcripts and with a similar level of coverage as those observed in fresh cell lines samples. Finally, as compared to SeqCap-lncRNA, the CoLong design enriched a higher number of currently annotated lncRNAs and, importantly, the CoLong design detected a significantly higher number of funcional lncRNAs than the SeqCap-lncRNA design (Figure 1B). Overall, we found a high correlation between the expression levels of lncRNAs across replicates (median Pearson correlation = 0.97), and before and after enrichment (Pearson correlation was 0.87 and 0.75 for polyA and total RNA, respectively), thereby underscoring the validity of this approach for the assessment of transcriptional differences. All together these results show that our approach efficiently enriched lncRNAs from fresh and FFPE samples, while preserving their relative abundance.

We next evaluated differences in the expression levels of lncRNAs enriched from fresh frozen (FF) biopsies (i.e., stored at −80 °C after surgery) and FFPE tissues in non-tumor samples of seven individuals for whom both types of samples were available. In all cases, we used total RNA to prepare the libraries and enriched for lncRNAs using the CoLong kit (Roche NimbleGen, Inc., Madison, WI, USA) before sequencing. Our results showed a high correlation of expression of lncRNAs between technical replicates (0.99) and, in agreement with earlier results for mRNAs, between FF and FFPE samples (0.85). However, FF and FFPE samples clustered separately (Figure 2). We found that shorter lncRNAs and those with higher GC content tended to show a lower expression in FFPE than in FF samples, which may reflect higher intrinsic degradation in the former type of preservation (Appendix A). Thus, although our results confirmed earlier findings of overall good agreement between FF and FFPE samples, they also pointed to potential differences, and thus a need to cross-validate FFPE results on other types of samples.

### 2.3. Identification of Differentially Expressed lncRNAs in CRC

We next set out to find lncRNAs that are differentially expressed (DE) in CRC tumors by applying our validated methodology (CoLong design) to FFPE samples. To detect DE lncRNAs, we used samples from 35 individuals with stage II CRC, which passed quality filters and had paired samples from tumoral tissue and the surrounding healthy tissue—referred to as normal hereafter. Different clustering approaches showed that expression profiles of lncRNAs separated samples in function of the disease condition, with an additional effect of sex (Figure 3).

Then, using a multi-factor design that considers ‘individual’ and ‘condition’ as independent variables (see Section 3), we identified 379 lncRNAs out of the 7812 lncRNAs included in the final dataset that were DE between normal and tumoral samples, with 48.8% and 51.2% of them being up- and down-regulated respectively (Figure 4A, Appendix A). Importantly, most of the samples showed the same directionality, as indicated by lines connecting normal and tumoral samples for a given individual (Figure 4B).

To compare the results of the enrichment-based FFPE procedure with those of the non-enrichment FF procedure, we downloaded RNAseq data from 40 matched pairs of samples from a CRC study available in The Cancer Genome Atlas (TCGA) [49] (Appendix A). Using the same procedure on this dataset as that described above, we identified 247 DE lncRNAs between normal and tumoral samples (Appendix A). Despite the different cohorts, nature of the samples, and protocols, we found that the comparisons had a significant overlap of 130 DE lncRNAs, which were similarly up- or down-regulated in the two studies. Of note, the enrichment-based FFPE dataset found a higher number of DE lncRNAs as compared to the FF study (379 vs. 247) and missed a lower fraction of DE lncRNA from the non-enriched dataset (117 out of 247) than the reverse comparison (249 out of 379). This observation suggests that differences are partly due to a greater ability of the enrichment-based approach to detect DE lncRNAs.

We examined whether DE lncRNAs identified exclusively in one of the two studies shared characteristics that may explain these discrepancies (Appendix A). In this regard, lncRNAs exclusively detected as DE in FFPE had lower expression levels, particularly in TCGA measurements, thereby suggesting that the absence of lncRNA enrichment in the TCGA study may have limited the analyses of lncRNAs with low expression (Appendix A). This notion is further supported by the observation that lncRNAs that were DE exclusively in the FFPE dataset showed lower expression values in data mined from several public repositories (Human Body Map, Array Express, ENCODE, etc) included in LnCompare [50] (see Section 3 and Appendix A).

We next assessed further characteristics of the lncRNAs detected as DE by each of the approaches and the overlap of the two. All three lists of DE lncRNA were enriched in functional lncRNAs (10% for FFPE, 13.7% for TCGA, and 17.7% for the overlap) compared to 3.2% functional lncRNAs in the CoLong design (*p*-value ≤ 1 × 10^−9^ after the hypergeometric test in all cases; see Section 3 and Appendix A). The three lists were also enriched in genes known to be involved in CRC (9% for FFPE, 13% for TCGA, and 17.7% for the overlap; *p*-value < 2 × 10^−10^ hypergeometric test in all cases, Appendix A). Similarly, we found that few DE lncRNAs had coding potential (5% of FFPE, 6.5% of TCGA and 5.4% of overlap), being candidates to encode for small peptides. We found that 43%, 39%, and 43% of DE lncRNAs in the FFPE, TCGA, and overlap lists, respectively, have evolutionarily conserved splice sites, which was similar for the bulk of all targeted genes (40.5%). The *FEZF1-AS1* gene, which was DE in the three lists, contains an ultraconserved element (uc232), which has been proposed to be involved in cancer [51]. However, its location overlaps with a protein-coding gene involved in cancer (*CADPS*). Finally, when comparing other characteristics of the lists of DE lncRNAs with GENCODE v24 lncRNAs [3], we found that the three lists were significantly enriched in functionally validated and disease associated lncRNAs, lncRNAs containing retroviral elements repeats (ERVL-MaLR,ERV1), lncRNAs genes with longer total and exonic lengths, lncRNAs with protein-coding genes upstream in the same strand, and with lower distances to the closest protein coding gene (adjusted *p*-value < 0.05, Appendix A). We searched for experimentally validated interactions of DE lncRNAs in the NPInter v4.0 database [52], and found that 82, 62, and 29 lncRNAs in the FFPE, TCGA, and overlap lists, respectively, interacted with RNAs or proteins (Appendix A). We performed a functional enrichment analysis of the set of interacting proteins and found significant enrichment in several biological processes (adjusted *p*-value < 0.05), which are summarized in Appendix A and Appendix A. Interestingly, some of the genes with interactions had no functional annotation in the databases, for instance, *ENSG00000231881* and *ENSG00000233858*). For *ENSG00000231881*, we found a recent study suggesting that it regulates the VEGF signaling pathway by binding to miR-133b, thereby promoting metastasis in CRC cells [52,53]. This notion is consistent with our results showing that *ENSG00000231881* is up-regulated in CRC tumors. Regarding *ENSG00000233858*, we found that it is downregulated in tumoral samples. This observation contrasts with a recent study that found that this gene was highly expressed in metastatic breast cancer, and associated it with poor prognosis [54]. That study concluded that *ENSG00000233858* and *UCA1* (*ENSG00000214049*) could cooperatively upregulate the expression of the *Slug* gene, which has been associated with poor prognosis and survival in CRC [55,56,57]. Interestingly, *UCA1* is significantly up-regulated in tumors in our analysis. Thus, the interplay of *UCA1* and *ENSG00000233858* is complex and may also be involved in CRC.

All together, our results underscore the power of a probe-based enrichment approach to assess DE lncRNAs in FFPE samples, and provide a candidate list of lncRNAs likely to be involved in CRC. In particularly, the 130 lncRNAs detected as DE in both the TCGA and FFPE datasets are strong candidates to be involved in CRC, as this set is significantly enriched in disease-related genes and in genes previously associated with this malignancy.

### 2.4. Validation of Gene Expression Data of a Subset of DE lncRNAs

To further validate the gene expression results obtained by our enrichment strategy on FFPE by an alternative approach, we used RT-qPCR on a subset of five lncRNAs significantly upregulated in tumoral samples: *ENSG00000247844 (CCAT1), ENSG00000226476 (LINC01748), ENSG00000259807 (AC009093.1), ENSG00000172965 (MIR4435-2HG), and ENSG00000227964 (LINC01429)*, of which the first three were also detected in the TCGA set. Given the general low level of expression of lncRNAs in FFPE samples, we selected the lncRNAs with the highest expression values among DE lncRNAs to ensure detection. In addition, we included two housekeeping genes as a reference in our validation: *ACTB* (a protein-coding gene) and *ENSG00000224635* (a lncRNA showing stable expression and thus used as internal control). We measured the Ct (cycle threshold) values of these transcripts in normal and tumoral FFPE samples and in the CRC cell line HCT116, as a control. We found no significant differences between normal and tumoral samples in the chosen housekeeping genes (Appendix A), thereby supporting their use as controls. In contrast, we found significantly higher Ct values (i.e., lower transcript amount) in normal as compared to tumoral FFPE samples (except for *ENSG00000259807*). This observation is consistent with the observed lower RNAseq expression values in the former (Figure 5).

In addition, the Cts for tumoral FFPE samples were higher than those of the CRC cell lines. This observation may be related to a lower quality of the RNA extracted from FFPE samples compared to that of fresh cell cultures or to differences in disease stage. Ct values showed a high dispersion, particularly in tumoral samples (Appendix A). Importantly, we found a significant negative correlation between RNA-Seq Log2(TPM) and RT-qPCR Delta Ct expression values not only in HCT116 cell line controls (R^2^ =  0.9, *p*  <  0.001), but also in FFPE samples (R^2^ =  0.27, *p*  <  0.001, Figure 5). When assessing this correlation in normal and tumoral samples separately, the significance was lost in the former (Appendix A), consistent with lower expression values.

### 2.5. Identification of CRC-Related Somatic Mutations in FFPE Samples

Genomic alterations are common in CRC and act as drivers of tumorigenesis [58]. Chromosomal instability, and variations in microsatellites and Cytosine-phosphate-Guanine (CpG) islands have been widely studied in the context of CRC, but little is known about somatic variants in lncRNAs. To date, only seven single nucleotide polymorphisms (SNPs) in lncRNAs have been associated with CRC [59]. Thus, we studied the presence of putative somatic mutations in lncRNAs using our RNAseq data from FFPE samples (see Section 3). We detected a higher number of putative somatic mutations in tumoral than in normal tissues (4995 and 1422, respectively). This finding is consistent with a previous study that concluded that CRC cells show a substantial increase in somatic mutation rate compared to normal colorectal cells [60]. Most somatic variants were present in a single individual (89.7% and 79.1% in tumoral and normal samples respectively), while variants shared by both tissues from an individual were commonly present in several individuals (Appendix A). Of note, we detected 184 somatic mutations shared by three or more individuals, and 11 of these were common and present in eight or more individuals (Appendix A). These eleven frequent CRC somatic variants affected eight lncRNAs (*ENSG00000226476, ENSG00000281406 (BLACAT1), ENSG00000248323 (LUCAT1), ENSG00000271762, ENSG00000282961 (PNRCR1), ENSG00000253929 (CASC21), ENSG00000245532 (NEAT1), ENSG00000265975*), of which the following five had a known role in CRC: *BLACAT1* [61,62]; *LUCAT1* [63]; *PNRCR1* [64]; *CASC21* [65]; and *NEAT1* [66]. In addition, four of these commonly mutated lncRNAs were DE in both the FFPE and TCGA datasets (*BLACAT1*, *LUCAT1*, *ENSG00000253929*, and *ENSG00000226476*). Our set of CRC somatic mutations includes one of the seven already described (rs7013433 in *CCAT1*), for which we found the same alternative allele reported in dbSNP [67]). Our dataset includes eight individuals who relapsed and we found that 21 of the CRC somatic variants showed higher frequency in relapsed as compared with non-relapsed (Appendix A). Four lncRNAs (*ENSG0000021403, ENSG00000261650, ENSG00000281406, G001643*) accumulated multiple such relapse-associated variants. These results suggest a prognostic value for those somatic variants and a possible role of these lncRNAs in cancer progression.

## 3. Material and Methods

### 3.1. Ethics Statement

This study was approved by the medical ethics boards of the Parc Taulí Hospital in Sabadell, Spain, code 2016510, received 01 February 2016. Tissue samples were obtained from FFPE blocks stored in the Pathology Department of the same hospital. Following the Spanish Law 14/2007 for Biomedical Research, informed consent was not necessary for these patients. According to the law, stored samples may be used for biomedical research without informed consent when obtaining informed consent is not possible or extremely difficult, provided that (i) the research performed is of general interest; (ii) there is no previous declaration of the patient against the use of the samples for research purposes; (iii) data confidentiality is guaranteed; and (iv) the Ethics Committee of the hospital evaluates and approves the study protocol.

### 3.2. Patients and Samples

Stage II FFPE-preserved colonic tumor tissue specimens and their surrounding non-cancerous normal mucosa were collected from 60 CRC patients who underwent surgical resection between 2003 and 2015 in the Department of Oncology at the Parc Taulí Hospital, Barcelona, Spain, without preoperative chemotherapy or radiotherapy. All patients were followed-up for at least 3 years. Tumors were classified following the tumor-node-metastasis (TNM) staging system. Fresh colonic tissue and their matched FFPE samples of eight patients were also included in the study. Epidemiological and clinico-pathological data of the patients were taken from the medical records and reviewed by medical oncologists. Detailed patient information is documented in Appendix A. Finally, we included two CRC cell lines as controls, namely HCT116 (ATCC CCL-247) and HT29 (HTB-38) from the Eukaryotic Cell Line Repository, CRG, Barcelona, Spain.

### 3.3. RNA Isolation From FFPE, FF, and Cell Line Cultures

Total RNA was isolated from 120 FFPE tissue specimens with the High Pure RNA Paraffin kit (Roche Diagnostics GmbH Mannheim, Germany, REF 03270289001). All reagents subsequently mentioned are from the High Pure RNA Paraffin kit if not specified otherwise. We performed two independent RNA isolations from each block, one for sequencing experiments and the other for RT-qPCR. It allowed us to perform all the experiments with fresh RNA, preventing RNA degradation and providing replicates of the biological source. From each FFPE block, two to three 10 µm sections were cut and placed in a 1.5 mL reaction tube. Samples were deparaffinized by adding 800–1000 µL of xylene (Merck KGaA, Darmstadt, Germany). After deparaffinization, following a modified protocol (provided by Dr. Nikic, Roche Diagnostics GmbH, Mannheim, Germany), we added 100 µL of tissue lysis buffer, 16 µL of 10% SDS (Merck KGaA, Darmstadt, Germany and 40 µL of proteinase K at each deparaffinized section. After vortexing for a few seconds, samples were spun down and incubated at 85 °C for 30 min under shaking at 600 rpm. Vials were then cooled to <55 °C and 80 µL of proteinase K was added. Samples were then vortexed for some seconds, spun down briefly, and incubated at 55 °C for 30 min under shaking at 600 rpm. From this step onwards, we followed the manufacturer’s protocol of the High Pure RNA Paraffin Kit (Version 12), with only one modification in the DNase treatment: we added 100 µL of DNase working solution (90 µL elution buffer + 10 µL DNase) and incubated the samples for 15 min at room temperature before adding the wash buffer I. For CRC cell lines and fresh frozen (FF) tissue biopsies, total RNA was extracted using the PureLink RNA mini kit (Ambion, Thermo Fisher Scientific, Waltham, MA, USA) following the manufacturer’s protocol, starting with 10^5^–10^7^ cells and between 100–200 ng of a single FF sample, respectively. DNase I treatment was applied during FFPE RNA extraction and after the RNA isolation protocol for FF samples and CRC cell lines (at 37 °C for 30 min). The purity and concentration of RNA were determined using Nanodrop 1000 spectrophotometer (Thermo Fisher Scientific, Walthman, MA, USA) and Quantus™ fluorometer (Promega, Madison, WI, USA). RNA integrity was determined using Bioanalyzer 2100 with the RNA 6000 Pico kit (Agilent Technologies, Santa Clara, CA, USA). Given the highly degraded status of nucleic acids in FFPE samples, we considered two RNA quality parameters: the RNA integrity number (RIN) and the DV200 (the percentage of fragments >200 nucleotides). For comparison purposes of the enrichment performance strategy between total RNA and polyA RNA samples, poly A selection of total RNA from CRC cell lines was carried out using the Dynabeads mRNA purification kit (Ambion, Thermo Fisher Scientific, Waltham, MA, USA) and following manufacturer’s instructions.

### 3.4. Custom Design of Probes for lncRNA Enrichment

cDNA libraries prepared from total RNA were enriched for lncRNAs using SeqCap EZ Choice Enrichment kits (Roche NimbleGen, Inc., Madison, WI, USA) before sequencing. We selected a total of 7812 transcripts to design probes, which corresponded to the longest transcript for each gene. Selected transcripts were classified as annotated and functional. Annotated transcripts are non-coding transcripts that do not overlap protein coding genes from Ensembl release 86, which uses GRCh38.p7 human genome version [68]. The gene biotype of these transcripts is generally lncRNA, meaning that all transcripts from a given gene are classified as lncRNA, with few exceptions that correspond to other non-coding categories (Appendix A). To compile the functional set, we mined literature and databases to select lncRNAs with annotated functions or with a potential role in CRC. In particular, we included 252 genes, comprising genes involved in CRC according to miTranscriptome [9], the Lnc2Cancer database [47], a recent literature review [29], and lncRNAs in lncRNAdb 2.0 [48], the latter being to date the largest database with experimentally curated functional lncRNAs (Appendix A). Second, we designed the probe sequences for the selected regions following the manufacturer’s instructions. This design is hereafter named CoLong.

### 3.5. Library Preparation, lncRNA Enrichment and Sequencing

A total of 408 cDNA libraries (corresponding to 136 samples) were prepared with the Illumina Truseq stranded Total RNA library preparation kit (Illumina, Inc., San Diego, CA, USA), following the manufacturer’s instructions, and starting from 100–150 ng of total RNA. Three technical replicates of library preparation were done per each sample. Final libraries were quantified by qPCR (KAPA library quantification kit, Roche, Basel, Switzerland) and the quality and size of the libraries were assessed using the Bioanalyzer 2100 and the DNA 1000 kit (Agilent Technologies, Santa Clara, CA, USA).

For comparison purposes, two distinct lncRNAs enrichment kits were used with the CRC cell lines: the commercial SeqCap lncRNA enrichment kit (SeqCap-lncRNA, onwards, Roche NimbleGen, Inc., Madison, WI, USA) and the above mentioned custom-designed CoLong, developed on the SeqCap RNA Choice XL enrichment kit (Roche NimbleGen, Inc., Madison, WI, USA). For the remaining samples (both the FFPE and FF biopsies), the lncRNA enrichment was done only with CoLong. Before using the NimbleGen SeqCap RNA enrichment system (Developer Kit, Roche NimbleGen, Inc., Madison, WI, USA), we prepared the multiplex cDNA sample library pools, a mixture of equal amounts (by mass) of each of the libraries to obtain one pool with a combined mass of 1 μg. In total 36 pools of 8 multiplexed libraries (125 ng each) and 4 pools of 16 multiplexed libraries (with an equimolar quantity of 62.5 ng each) were prepared. Then, this 1 μg of each multiplex cDNA sample library was hybridized to a total of 3 commercial and 37 custom probe-enrichment kits following the manufacturer instructions. All reagents subsequently mentioned are from the NimbleGen SeqCap RNA Developer enrichment kit if not specified otherwise. Briefly, each multiplex cDNA sample library pool (1 μg) was mixed with 5 μg of COT human DNA and 2000 pmol of the corresponding multiplex hybridization enhancing oligo pool (to prevent hybridization between adapter sequences). After drying this mixture in a DNA vacuum concentrator at 60 °C, the following reagents were added: 7.5 μL of 2 times concentrated hybridization buffer and 3 μL of hybridization component A. Samples were vortexed for 10 s, centrifuged at maximum speed for 10 s, and then left at 95 °C for 10 min to denature the cDNA. After a short centrifugation at maximum speed for 10 s, the mixture was transferred to a 4.5 μL aliquot of the SeqCap RNA probe pool previously prepared in a 0.2 mL PCR tube, vortexed for 3 s, and centrifuged at maximum speed for 10 s more. Finally, the mixture was incubated in a thermocycler at 47 °C for 20 h (with the thermocycler lid set at 57 °C). After the hybridization step, samples were washed and the captured multiplex cDNA samples were recovered from the mixture with SeqCap streptavidin beads and amplified following the manufacturer’s instructions. PCR products were purified with AMPure XP beads (Beckman Coulter Inc., Brea, CA, USA).

The quality of the enriched pools was assessed with a bioanalyzer DNA 1000 chip (Agilent Technologies, Santa Clara, CA, USA). Enriched pools of libraries were run in an Illumina Hiseq 2500 (Illumina, Inc., San Diego, CA, USA), using a single 50 nucleotide read configuration. Two pools were run paired-end with a read length of 125 bp and analyzed accordingly (Pool 14: samples 51N, 51T, 52N, 52T, 53N, 53T, 54N, 54T, and Pool 15: samples 55N, 55T, 56N, 56T, 47N, 47T, 48N, 48T). For 8 multiplexed pools, we ran a total of 3 pools per lane except for the initial pool of control samples (HCT116 and HT29), which were run in 1/9th of a lane. For 16 multiplexed pools, we ran a total of 2 pools per lane.

### 3.6. Quality Control, Pseudo-Mapping and Gene Abundance Quantification

We used FastQC v0.11.6 (https://www.bioinformatics.babraham.ac.uk/projects/fastqc/) [69] and Multiqc v1.0 [70] to perform quality control of raw sequencing data. For the samples with paired-end 125 bp reads, we performed adapter trimming with Trimmomatic v0.36 [71] with TruSeq3 paired-end adapters using ILLUMINACLIP: 2:30:10 parameters and the minimum read length of 50 bp. We compared the performance of the non-enriched, and SeqCap-lncRNA- and CoLong-enriched samples by mapping the RNA-Seq reads to the primary human genome assembly GRCh38 [72] and providing a genome annotation (gtf) file containing all protein coding genes and studied lncRNAs in Ensembl r86 using STAR v2.5.3a splice-junction aware mapping software [73].

To quantify the expression levels of the studied lncRNAs in the FF and FFPE samples, we used the pseudoalignment approach implemented in Salmon v0.12.0 [74]. First, we extracted the transcriptome from the top-level human genome assembly GRCh38 from the Ensembl database (last accessed 16/02/2018, [72]) by RSEM prepare-reference v1.2.28 [75] using the genome annotation (gtf) file containing all protein coding genes and lncRNAs studied, as recommended in [76]. Next, we performed pseudoalignment of RNA-Seq data to the obtained custom transcriptome via Salmon using--dumpEq --incompatPrior 0 --gcBias --validateMappings options [74].

### 3.7. Differential Expression Analysis

For differential gene expression analysis, we used DESeq2 v1.22.2 [77]. In brief, we imported the gene-level abundance estimates obtained from Salmon using tximport v1.10.1 and txOut = T option (https://bioconductor.org/packages/release/bioc/html/tximport.html). Our design included individual and condition (normal and tumoral samples) as independent variables. The inclusion of individual as an independent variable accounts for the paired nature of the data and the sex. We pre-filtered the data by removing genes that had no counts and those for which more than 75% of the samples had normalized counts lower than 1. We then combined the counts from the three technical replicates using the collapseReplicates function and, after inspection of the principal component analysis (PCA) plot, we discarded those samples that were not correctly clustered. PCA was computed using transformed count data obtained using the vst method. For the purpose of visualization (but not to infer the number of differentially expressed genes), we removed batch effects using the removeBatchEffect function of the limma package v3.38.3. To assess the overall similarity between filtered samples, we calculated the Euclidean distance between samples using the vst-transformed data and the dist function from R. A heatmap was obtained using heatmap.2 function after providing a hierarchical clustering that was computed using the Euclidean distances and hclust function from R. We ran differential expression analysis using the deseq function and we extracted the results after filtering out genes in which more than 75% of the samples had normalized counts lower than 1 (note than some samples were filtered out in the quality control step), with an adjusted *p*-value lower than 0.05 and log2 fold-change ranging between −1 and 1. We have indicated also in Appendix A whether the log2 fold change of a given lncRNA is outside the −2 to 2 range. All these analyses were performed using R v3.5.2.

### 3.8. Gene Expression Quantification of TCGA Samples

We downloaded a total of 85 matched samples (normal and tumoral) from 40 individuals, 4 of them including replicas for tumoral samples from the Cancer Genome Atlas (TCGA) [49]. Controlled data (bam files) were downloaded through NCI Genomic Data Commons (GDC portal, accession number phs000178) [78]. We converted the bam files into fastq using the fastq function of samtools v. 1.9 [79] and used the pipeline described above for pseudo-mapping and gene abundance quantification. The PCA analysis of transformed count data (vst method) showed that samples were clustered by condition (normal, tumoral) but also by the type of library used (single end and paired end). To control for batch differences, we included the type of library in the design when performing the differential expression analysis, using the procedure described above.

### 3.9. Characterization of Differentially-Expressed lncRNAs

Enrichment in functional lncRNAs was assessed after a hypergeometric test using the Phyper function from R v.3.6.3. We used LnCompare [50] to compare the biological properties between the list of differentially expressed (DE) lncRNAs and the entire GENCODE v24 annotation as background. We searched for evolutionary conserved Splice sites using SpliceMap WebServer (http://splicemap.bioinf.uni-leipzig.de) [80]. We also identified ultraconserved elements by performing blast searches using an e-value cutoff of 1 × 10^−3^ using as a query sequences from ultraconserved elements, downloaded from https://users.soe.ucsc.edu/~jill/ultra.html on 10 May 2019. We computed the coding potential using CPC (http://cpc.gao-lab.org/) and CPAT (http://lilab.research.bcm.edu/cpat/, accessed 25 March 2020) [81,82]. We assessed overlaps with protein coding genes using a more recent annotation of the human genome (Ensembl r96). We assessed experimentally validated interactions using NPInter v4.0 db [52]. We subsequently performed a functional enrichment analysis of the interacting proteins using Fatigo implemented in Babelomics 5 [83,84] and then visualized the results using Revigo with default parameters [84].

### 3.10. SNP Calling

To perform variant calling, we first mapped the RNA-Seq reads to the human genome assembly GRCh38 from the Ensembl database [72] providing the same gene annotations (gtf) used for expression quantification, using STAR v2.5.3a splice-junction aware mapping software [73]. The bam files were sorted with samtools v1.9. Further variant calling was done by bcftools v1.8 [85], using mpileup function with max-depth 2000 option and default parameters. We normalized the variants in each VCF file by left-aligning indels and splitting multi-allelic calls, selected variants located in the enriched lncRNAs, and discarded variants with QUAL < 20. We then filtered out variants that were not consistently called in technical replicates of each normal and tumoral sample, by checking chromosome, position, reference, and alternative alleles. We then classified variants as unique for tumoral and normal samples (somatic mutations) and those present in both types of samples using an in-house python script.

### 3.11. Validation of lncRNA Expression by Real-Time Quantitative PCR (RT-qPCR)

We used RT-qPCR to validate gene expression results obtained by RNA-Seq. We selected five lncRNAs (*CCAT1, ENSG00000226476, ENSG00000259807, ENSG00000172965, LINC01429*) based on differences in expression between normal and tumoral samples showing high fold-change and base mean values (as calculated by DEseq2, being the mean number of normalized counts for all samples after normalizing for sequencing depth). We also selected two housekeeping genes: ACTB (protein-coding gene) and *ENSG00000224635* (internal control from the dataset). In brief, we isolated total RNA from 106 FFPE samples (corresponding to 53 normal samples and 53 tumor samples) using high pure RNA paraffin kit (Roche, Basel, Switzerland) and a control RNA isolated from CRC cell line HCT116. We quantified the total RNA (Quantus™ fluorometer, Promega, Madison, WI, USA) and verified the quality of the samples (bioanalyzer 2100 RNA nano Kit, Agilent Technologies, Santa Clara, CA, USA). cDNA synthesis was performed using the PrimeScript RT reagent kit (Takara Biotechnology) following the manufacturer’s instructions. The reverse transcriptase (RT) reaction contained 200 ng of total RNA template, 4 μL of 5× PrimeScript buffer, 1 μL of PrimeScript RT enzyme mix I, 1 μL of OligodT primer (50 μM), 4 μL of random 6-mers (100 μM), and nuclease free water to a total volume of 20 μL. Reactions were incubated in a thermocycler (Analytik Jena AG, Jena, Germany) for 15 min at 37 °C, followed by heat inactivation of RT for 5 s at 85 °C and a hold step at 4 °C. For each sample, we performed a total of two RT reactions, and then pooled them together, and one non-reverse transcribed control reaction (RT−). The probes for qPCR assays were designed with the PrimerQuest design tool from IDT (www.idtdna.com/SciTools) (Appendix A). All probes were intron-spanning, had an average amplicon size of 100 bp long, and were located in exons covered by RNA-Seq results. Each qPCR reaction contained 5 μL of Prime Time gene expression master mix, 0.5 μL of Prime Time qPCR assay (20×) (Integrated DNA Technologies, IDT, Newark, NJ, USA), and 1 μL of cDNA template in a final volume of 10 μL. Each sample was assayed in triplicate. Both water blanks and non-reverse transcribed RNA samples (RT-) were used as negative controls. Reactions were incubated in a 384-well optical plate (4titude, Wotton, UK) at 95 °C for 3 min, followed by 40 cycles at 95 °C for 5 s and 60 °C for 30 s, in a QuantStudio™ 7 flex real-time PCR system (Applied Biosystems, Thermo Fisher Scientific, Waltham, MAUSA).

Samples showing no amplification, samples with Ct values greater than 36 cycles, as well as replicates with deviations from the mean greater than 0.5 cycles were discarded. We then calculated the relative expression of each target lncRNA using the ‘Delta Ct method’ (Ct target—Ct housekeeping) [86]. To calculate the correlation among techniques, we compared gene expression values of RNA-Seq (expressed as log2(TPM)) and RT-qPCR (expressed as DeltaCt) for all the samples and genes assayed, considering the two housekeeping genes independently, by using Pearson correlation (stat_cor function from R).

## 4. Conclusions

Altogether, our results provide compelling evidence of the validity of a probe-enrichment approach to study lncRNA transcripts in various types of samples, including FFPE-preserved tissue. FFPE is commonly used in hospitals to preserve valuable material for further clinical and translational research. However, this material is prone to nucleic acid degradation, which hinders transcriptional studies, particularly of lncRNA. Our results show that the enrichment approach described herein preserves relative levels of lncRNA abundance, thereby enabling differential expression studies. In addition, they show that expression patterns obtained from matched FFPE and FF samples are highly correlated but distinct. This observation implies that cross-validation of studies on FFPE-preserved samples with other types of samples is recommendable. In addition to expression analyses, we show that our approach is useful for genotyping lncRNAs expressed in FFPE samples, thereby allowing the discovery of disease-associated somatic mutations. Our custom-designed CoLong, which included 252 lncRNAs previously shown to be involved in CRC, showed a higher specificity for lncRNAs with annotated functions, thus highlighting the relevance of careful probe design.

The use of the validated probe-based enrichment approach on a cohort of stage II CRC patients with matched tumor and normal FFPE tissues, identified 379 lncRNAs DE between the two types of tissues. We compared this result with the analysis of a publicly available dataset of FF samples. Despite the distinct cohorts, disease stage, and sample types of the two studies, the set of DE lncRNAs overlapped significantly. Our enrichment-based procedure detected a higher number of DE lncRNAs, particularly among those that have lower expression. The 130 lncRNAs detected as DE by the two datasets, is enriched in genes known to be involved in CRC: 17.7%, as compared to 9–13% among those detected in a single dataset and to a background of 3.2% of the lncRNAs included in our probe set. The remaining, non characterized lncRNAs are thus very strong candidates for being involved in CRC and should be prioritized in future research. In addition, we genotyped expressed lncRNAs using RNAseq data derived from probe-based enrichment of FFPE samples. We identified a catalogue of somatic mutations exclusively found in tumors, and quantified their prevalence in individuals that relapsed within three years after tumor resection. Our results expand the catalogue of CRC-associated lncRNA variants (from 11 known variants to 184 variants in our study) and identify lncRNA sequence variants that are particularly enriched in tumors that experienced a relapse.

In summary, our study provides the first evidence of the feasibility of studying differential expression and somatic mutations in lncRNAs using a probe-enrichment approach and FFPE samples. Our results indicate that numerous lncRNAs alter their expression and/or accumulate variants in CRC tumors, thereby highlighting the importance of this non-coding class of genes in CRC progression. Our catalogue of altered lncRNAs deserves attention in future studies aiming to discern the role of lncRNAs in CRC and other cancer types. The new methodology presented herein paves the way for the analysis of lncRNAs in large collections of preserved clinical samples available at many hospitals. The significance of this technical breakthrough extends well beyond clinical research in CRC and cancer.

## Figures and Tables

**Figure 1 cancers-12-02844-f001:**
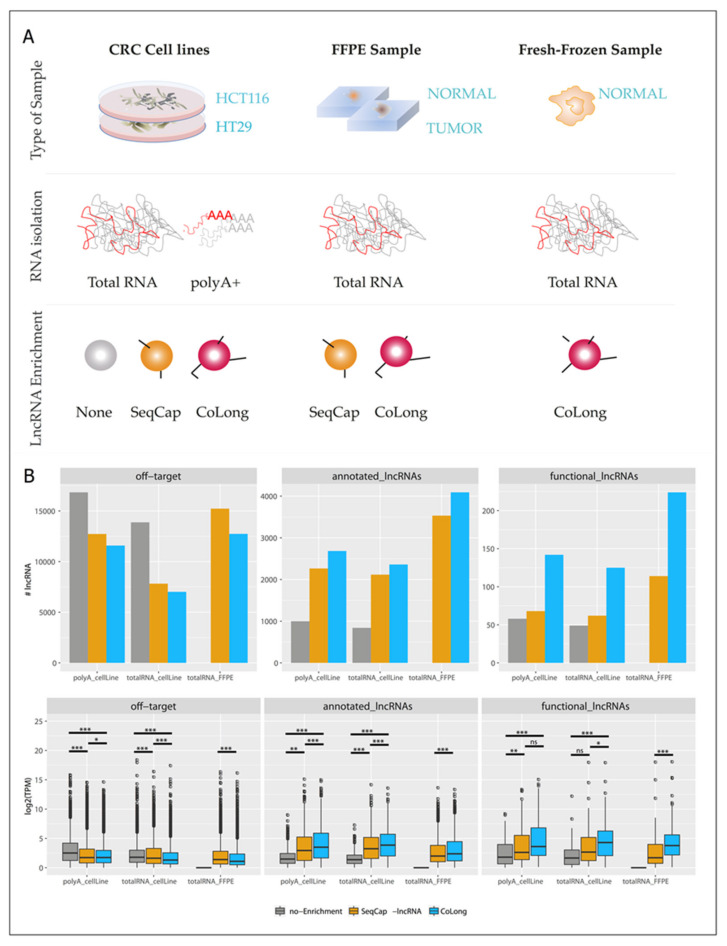
Validation of probe-set SeqCap enrichment with the CoLong design. (**A**) Scheme summarizing the experimental strategy undertaken to pursue the validation of the CoLong probe set. In brief, we used three types of sample source (fresh colorectal cancer (CRC) cell lines, formalin-fixed, paraffin-embedded (FFPE) samples from CRC patients (tumor and normal tissues), and fresh-frozen samples from normal tissue of candidates for colonoscopy. We isolated total RNA (and polyA+ selection in CRC cell lines) and compared the performance of lncRNA detection: in the absence of enrichment (none), with the SeqCap enrichment commercial design, and with the CoLong design. (**B**) Comparison of gene quantification between the two enrichment protocols (SeqCap-lncRNA [25] and CoLong (present study)) and without enrichment, using total RNA and polyA enrichment in CRC cell lines and FFPE samples. Top: number of lncRNAs with TPM > 1 detected in each condition; bottom: median log2(TPM) values for the replicates in each experiment. *P*-values were calculated using a Wilcoxon test: *p*-value > 0.05: not significant (ns), *p*-value < 0.05: *, *p*-value < 0.01: **, *p*-value < 0.001: ***.

**Figure 2 cancers-12-02844-f002:**
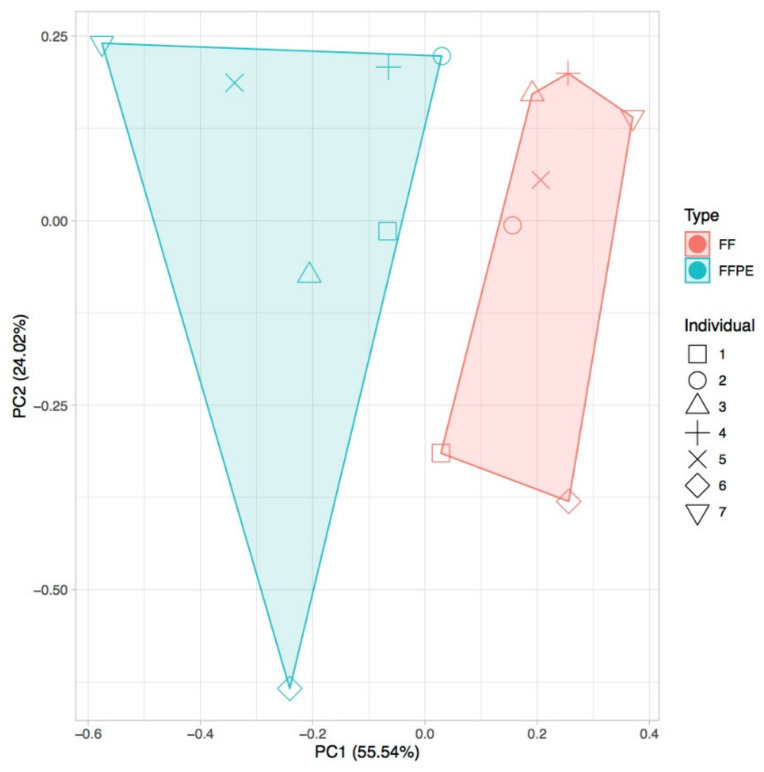
Principal component analysis (PCA) for fresh frozen (FF) and formalin-fixed, paraffin-embedded (FFPE) matched samples from seven individuals, using transcripts-per-million (TPM) values of gene expression (average for all replicates) and discarding genes with TPM < 1.

**Figure 3 cancers-12-02844-f003:**
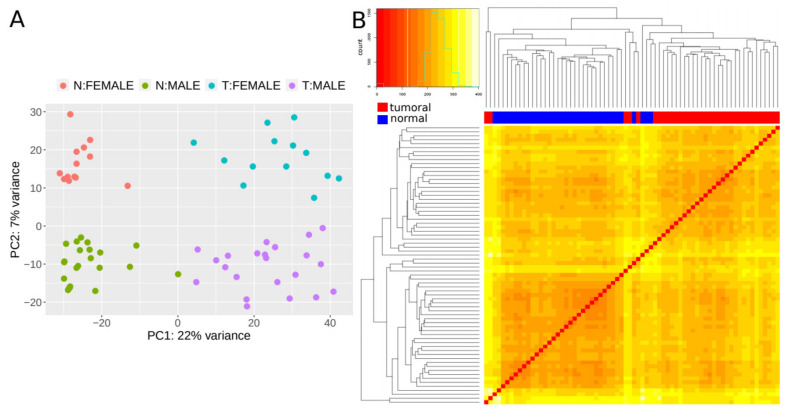
Expression patterns of lncRNAs. (**A**) Principal component analysis (PCA) plot based on transformed count data after variance stabilizing transformation. T and N indicate tumoral and normal samples, respectively. (**B**) Heatmap showing the overall similarity between samples based on a hierarchical clustering of the Euclidean distance between samples using the variance stabilizing transformation (vst-transformed) data. Red and blue bars at the top of the columns designate tumoral and normal samples, respectively.

**Figure 4 cancers-12-02844-f004:**
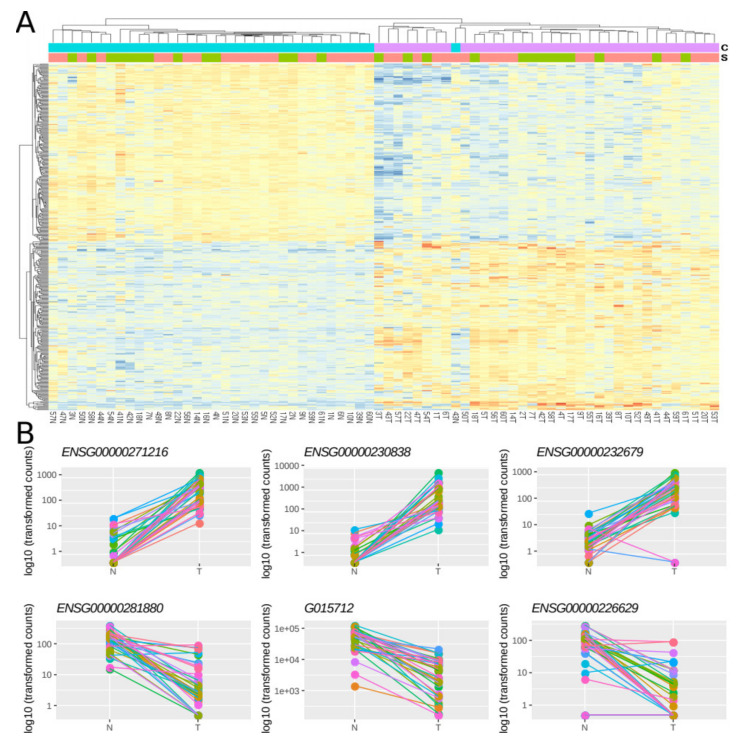
Differentially expressed lncRNAs in FFPE CRC stage II samples. (**A**) Heatmap showing gene expression from the 379 DE lncRNAs. Color gradient ranges from low expressed (blue) to highly expressed (red). (**B**) Dot plots showing normalized count data in a logarithmic scale of normal and tumoral samples for the three top up- and down-DE lncRNAs (Appendix A). Horizontal bars on top show condition (blue for normal, pink for tumoral) and sex (green for female and red for male). Each individual is represented with a different color, lines group paired samples from the same individual and T and N indicate tumoral and normal samples, respectively.

**Figure 5 cancers-12-02844-f005:**
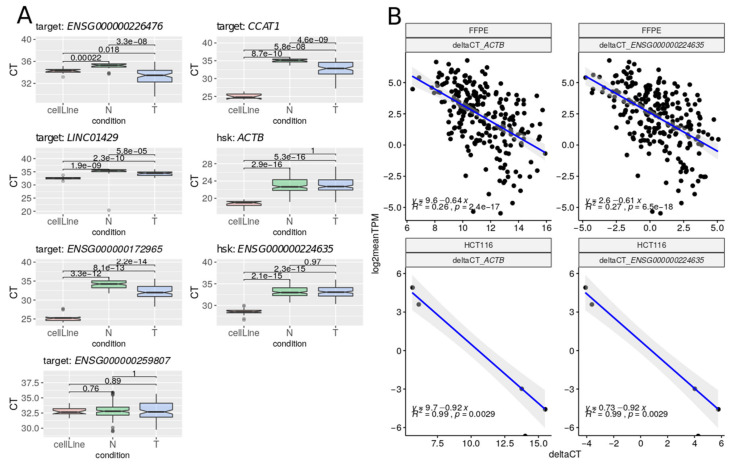
(**A**) Ct (cycle threshold) levels for each selected transcript in normal (N), tumoral (T), and control (CRC cell line HCT116) samples. (**B**) Gene expression levels between RNA-Seq (log2(TPM)) and RT-qPCR (Delta Ct values), considering the two housekeeping genes independently. Pearson correlations were estimated using stat_cor function from R v3.6.3.

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
