# Peer review of "Target Enrichment Enables the Discovery of lncRNAs with Somatic Mutations or Altered Expression in Paraffin-Embedded Colorectal Cancer Samples"

_cancers, 2020, doi:10.3390/cancers12102844_

Round 1
Reviewer 1 Report
In this manuscript authors describe a probe-based strategy to profile lncRNA expression in Formalin-fixed, paraffin-embedded (FFPE) tissues and determine its improvements over existing technologies. Manuscript is well written and methods are sufficiently described. The authors have well taken care of the parameters that we generally look for while filtering the gene sets (for example, checking for lncRNAs with an adjusted p value of <0.05 and log2 fold change of -1 to 1). Although a value from -2 to 2 is preferred, given the overall low expression of the lncRNAs in these samples I would consider it. Since they have used the standard R packages for analyzing the DE lncRNAs I would assume that the results are reliable. It appears from the data that their enrichment method can be used to mine transcriptomics related data from FFPE samples. The only revisions I ask is that the font size of the figures, especially the heatmap font resolution needs to be improved and a better explanation of why only stage II samples were taken needs to be given.
Author Response
Response:
We thank the reviewer for his/her comments.
We understand the concerns regarding the log2 fold change threshold. We have added an additional column in Table S1 and Table S2 indicating whether the log2 fold change of a given lncRNA is outside the -2 to 2 range. Importantly, most of the differentially expressed lncRNAs are outside that range and thus would pass this more stringent threshold (92.6% of FFPE samples and 94.3% of TCGA samples). Thus, we consider that the overall results that are described in the manuscript would remain.
We have improved the resolution of all the figures and modified the font size when possible. In the case of the heatmap, we have opted for removing the sample names from the figure, as with so many samples the names will not be readable, and they are not relevant for the purpose of that figure.
With regards to the choice for stage II colon samples. We focused on this stage as it is particularly difficult to predict prognosis and select the more suitable therapy. The value of adjuvant chemotherapy for patients with stage II disease is controversial despite multiple clinical trials and meta-analyses. Questions remain not only about which patients will benefit from treatment but also what chemotherapy to use if adjuvant chemotherapy is recommended. ASCO and ESMO guidelines recommended considering adjuvant chemotherapy for patients with stage II colon cancer with high-risk features and emphasize the importance of discussing the very real risks of chemotherapy, including permanent neuropathy and even fatal complications from chemotherapy, balanced against the potential minimal improvement in OS. Current National Comprehensive Cancer Network (NCCN) guidelines for treatment of stage II colon cancer suggest a discussion with the patient regarding the risks and benefits of adjuvant chemotherapy in addition to clinical risk assessment (using number of lymph nodes analyzed, tumor stage, etc.) As a result, there is no standardized approach to determining which (if any) stage II colon cancer patients should receive adjuvant chemotherapy. And, in practice many patients with stage II colon cancer without high-risk features receive adjuvant chemotherapy. The development of molecular markers that predict clinical outcome or response to therapy in stage II colon cancer is an important tool that could give clinicians added information in discussions regarding the role of adjuvant chemotherapy. We have better explained our choice in the current version of the manuscript, line 90.
Reviewer 2 Report
The authors designed and tested a custom lncRNA enrichment probe set. They focused on samples from stage II CRC patients, for which novel prognostic tools are needed and they identified 379 lncRNAs differentially expressed
Major points:
- Please explain better the choice to consider stqge II CRC and not stage III( more aggressive )
- Figure 1 b the authors should indicate a p-value
- In material and method the authors declare that the comparison if 40 normal vs 45 CRC. Is it right?
In the section 2.3 the authors declare to use 35 CRC for differential expression analysis. Please correct or explain better.
- Figure 2 . please indicate the two panels left and right as A) and B) in the figure.
- Figure 2b the authors should indicate a legend where they explain what the blue and red bars represent. In addition, they should improve the resolution of the legend 2B of the heatmap. I can’t read the numbers
- The authors identified 379 DE lncRNAs. But the authors should specify how many total lncRNAs are there?
- The authors should explain better the figure 3 in the main text. In addition the resolution is low
- Figure 4 the authors should indicate the p-values
Author Response
Reviewer 2:
The authors designed and tested a custom lncRNA enrichment probe set. They focused on samples from stage II CRC patients, for which novel prognostic tools are needed and they identified 379 lncRNAs differentially expressed
Major points:
- Please explain better the choice to consider stqge II CRC and not stage III( more aggressive )
Response:
We thank the reviewer for the opportunity to clarify better the selection of the samples. In this study, we aim to develop an efficient method to identify and characterize lncRNAs, and to explore its potential as biomarkers for CRC prognosis. In this regard, we selected specifically stage II colon cancer samples (in which the lymph nodes have not yet been affected), since at this stage it is particularly difficult to predict prognosis and select the more suitable therapy.
The value of adjuvant chemotherapy for patients with stage II disease is controversial despite multiple clinical trials and meta-analyses. Questions remain not only about which patients will benefit from treatment but also what chemotherapy to use if adjuvant chemotherapy is recommended. ASCO and ESMO guidelines recommended considering adjuvant chemotherapy for patients with stage II colon cancer with high-risk features and emphasize the importance of discussing the very real risks of chemotherapy, including permanent neuropathy and even fatal complications from chemotherapy, balanced against the potential minimal improvement in OS. Current National Comprehensive Cancer Network (NCCN) guidelines for treatment of stage II colon cancer suggest a discussion with the patient regarding the risks and benefits of adjuvant chemotherapy in addition to clinical risk assessment (using number of lymph nodes analyzed, tumor stage, etc.) As a result, there is no standardized approach to determining which (if any) stage II colon cancer patients should receive adjuvant chemotherapy. And, in practice many patients with stage II colon cancer without high-risk features receive adjuvant chemotherapy.
The development of molecular markers that predict clinical outcome or response to therapy in stage II colon cancer is an important tool that could give clinicians added information in discussions regarding the role of adjuvant chemotherapy. We have better explained our choice in the current version of the manuscript, line 90
Reviewer 2:
- Figure 1 b the authors should indicate a p-value
Response:
The p-value is now indicated in Figure 1b.
Reviewer 2:
- In material and method the authors declare that the comparison if 40 normal vs 45 CRC. Is it right?
In the section 2.3 the authors declare to use 35 CRC for differential expression analysis. Please correct or explain better.
Response:
The numbers are correct since they refer to two different data sets. Section 2.3 refers to FFPE samples and in line 434 we refer to TCGA samples. With regards to TCGA samples we have better explained in the text. In line 466: 85 matched samples (normal and tumoral) from 40 individuals, 4 of them including replicas for tumoral samples.
Reviewer 2:
- Figure 2 . please indicate the two panels left and right as A) and B) in the figure.
Response:
The panels are now referred as A) and B)
Reviewer 2:
- Figure 2b the authors should indicate a legend where they explain what the blue and red bars represent. In addition, they should improve the resolution of the legend 2B of the heatmap. I can’t read the numbers
Response:
The legend already indicated that red and blue colors designate tumoral and normal samples, respectively, but we have clarified the sentence as "Red and blue bars at the top of the columns designate tumoral and normal samples, respectively." As explained in the response to reviewer 1, the sample names are not relevant for this figure which simply wants to show the clustering per sample type, so we have removed the labels to improve clarity.
Reviewer 2:
- The authors identified 379 DE lncRNAs. But the authors should specify how many total lncRNAs are there?
Response:
In the section 2.1 (line 112-14) we specify that: The final dataset comprised 7,812 lncRNA genes, of which 216 can be considered as CRC-related on the basis of previous knowledge. But, to improve clarity we have also added this information later in section 2.3 (line 186)
Reviewer 2:
- The authors should explain better the figure 3 in the main text. In addition the resolution is low
Response:
We have now improved the resolution of the Figure and now better explain Figure 3 in the main text: …”we identified 379 lncRNAs that were differentially expressed between normal and tumoral samples, being 48.8% and 51.2% of them up- and down-regulated respectively (Figure 3A, Supplementary Table S1). Importantly, most of the samples showed the same directionality, as indicated by lines connecting normal and tumoral samples for a given individual (Figure 3B).”
Reviewer 2:
- Figure 4 the authors should indicate the p-values
Response:
The p-value is now indicated in Figure 4.
Reviewer 3 Report
In this manuscript, the authors designed a custom lncRNA enrichment probe set and tested its potential of targeted lncRNAs enrichment from FFPE samples, which can greatly expand the availability of material for cancer studies.
The following concerns are raised:
- Line 142 is not concise and clear, may rephrase it to underscore the functional lncRNA group
- Line 322-323, why the authors perform two independent isolations, one for sequencing, the other for RT- qPCR, rather than only one isolation, half/majority of the resultant RNA for sequencing, the rest for RT-qPCR?
- In line 148-159, for the seven biological replicates for both types of tissues, more analysis should be included to support the validity of the custom lncRNA enrichment method, such as annotated and off-target lncRNA.
- In line 152, “between replicates” should be clear, not technical replicates.
- Supplementary figure 1 may be considered as the main figure to demonstrate the validity of CoLong method in FFPE samples.
- In line 363, the authors indicated that “Three technical replicates of library preparation were done per each sample”. However, in Figure 1B, the top panel has no median values as the bottom panel
- Based on Figure 1B, it shows that both methods SeqCap-lncRNA and CoLong give rise to a much higher number of off-target lncRNA, is this common for currently available lncRNA enrichment methods?
Author Response
Reviewer 3
In this manuscript, the authors designed a custom lncRNA enrichment probe set and tested its potential of targeted lncRNAs enrichment from FFPE samples, which can greatly expand the availability of material for cancer studies.
The following concerns are raised:
- Line 142 is not concise and clear, may rephrase it to underscore the functional lncRNA group
Response:
We agree with the reviewer and we have rephrased for clarity (line 151): “Importantly, the CoLong design detected a significantly higher number of functional lncRNAs than the SeqCap-lncRNA design (Figure 1B)”.
Reviewer 3
- Line 322-323, why the authors perform two independent isolations, one for sequencing, the other for RT- qPCR, rather than only one isolation, half/majority of the resultant RNA for sequencing, the rest for RT-qPCR?
Response:
We thank the reviewer 3 for the comment, and modified the text accordingly to specify the reason for performing two independent isolations of RNA (line 351): "We performed two independent RNA isolations from each block, one for RNA sequencing and the other one for RT-qPCR. It allowed us to perform all the experiments with fresh RNA, preventing RNA degradation and providing replicates of the biological source."
Reviewer 3
- In line 148-159, for the seven biological replicates for both types of tissues, more analysis should be included to support the validity of the custom lncRNA enrichment method, such as annotated and off-target lncRNA.
Response:
We thank the reviewer for raising this point. We have now included the analysis of the differentially expressed off-target genes, which supports the results found for functional and annotated lncRNAs. We show this analysis in the new Supplementary Figure S1 (former Supplementary Figure S2)
Reviewer 3
- In line 152, “between replicates” should be clear, not technical replicates.
Response:
We thank the reviewer for pointing this out, we have clarified the sentence stating that these are technical replicates. Now the sentence reads as follows: “Our results showed a high correlation of expression of lncRNAs between technical replicates (0.99)”
Reviewer 3
- Supplementary figure 1 may be considered as the main figure to demonstrate the validity of CoLong method in FFPE samples.
Response:
As suggested by the reviewer we have included Supplementary Figure 1 as main Figure 2.
Reviewer 3
- In line 363, the authors indicated that “Three technical replicates of library preparation were done per each sample”. However, in Figure 1B, the top panel has no median values as the bottom panel
Response:
We thank the reviewer for pointing this out. We have rewritten the Figure 1 footnote for clarity: “B) Comparison of gene quantification between the two enrichment protocols (SeqCap-lncRNA and CoLong) and without enrichment, using total RNA and polyA enrichment in CRC cell lines and FFPE samples. Top: number of lncRNAs with TPM>1 detected in each condition; bottom: median log2(TPM) values for the replicates in each experiment. P-values were calculated using a wilcoxon test: p-value>0.05: ns, p-value < 0.05: *, p-value < 0.01: **, p-value<0.001: ***”
Reviewer 3
- Based on Figure 1B, it shows that both methods SeqCap-lncRNA and CoLong give rise to a much higher number of off-target lncRNA, is this common for currently available lncRNA enrichment methods?
Response:
Enrichment methods enrich for the target sequences, but cannot get completely rid of off-target molecules, therefore a high number of off-target molecules is expected by any enrichment method. What is of relevance is the fraction of reads recovered for the target molecules as compared for the non-target, which is seen in the graph showing the Log1(TPM) values, both enrichment methods (but CoLONG) more so, effectively enrich the target molecules.
Reviewer 4 Report
The manuscript is well written and interesting. Guzmán et al discussed an approach that unlocks the study of lncRNAs in FFPE samples, thus enabling the retrospective use of abundant, well-documented material available in hospital biobanks. Overall this manuscript has merit and the outcome of this study also has novelty.
I have only one minor comment- Please provide a schematic representation of this study.
Author Response
Response:
We thank the reviewer for his/her comments, we have added a schematic representation of the study as a Supplementary Figure S1.
Round 2
Reviewer 2 Report
tha manuscript has been significanlty improved